# Comparative Study of Classical and Alternative Therapy in Dogs with Allergies

**DOI:** 10.3390/ani12141832

**Published:** 2022-07-19

**Authors:** Alena Micháľová, Martina Takáčová, Martina Karasová, Lukáš Kunay, Simona Grelová, Mária Fialkovičová

**Affiliations:** Small Animal Clinic, Department of Internal Diseases, University of Veterinary Medicine and Pharmacy, 04181 Košice, Slovakia; martina.taka@gmail.com (M.T.); martina.karasova@uvlf.sk (M.K.); kunay.luk@gmail.com (L.K.); sgrelova@gmail.com (S.G.); maria.fialkovicova@uvlf.sk (M.F.)

**Keywords:** allergy, traditional Chinese medicine, acupuncture, phytotherapy, nutrition, conventional Western medicine therapy

## Abstract

**Simple Summary:**

Acupuncture, phytotherapy, and nutrition are part of traditional Chinese medicine, which has been used for literally hundreds to a few thousand years. These traditional therapeutic methods can effectively diagnose and treat acute and chronic diseases and can be used as a primary or complementary therapy. Used properly, these alternatives are safe and without side effects. Allergy is currently a very common diagnosis affecting dogs. Conventional Western medicine can treat symptoms but often does not identify and resolve the underlying problem. This comparative study was focused on the application of alternative and conventional medicine in allergic conditions in dogs, which were divided into two groups, where the effectiveness of both types of treatment was compared.

**Abstract:**

Allergy is a malfunction of the immune system that causes an inappropriate reaction to normally harmless substances known as allergens, such as food components, pollen, parasites, mites, medication, etc. It is very important to make a correct diagnosis, to identify and to eliminate the offending allergen from the body, and provide control and long-term management to achieve a comfortable life for the animal. In the case of highly intensive pruritus, drugs such as glucocorticoids, antihistamines, and Janus kinase inhibitors are generally administered. Unfortunately, common drugs are not always able to resolve the problem. This comparative clinical-outcomes study focused on the application of alternatives, where a combination of acupuncture with phytotherapy and nutrition was applied. These traditional methods do not affect the body only symptomatologically; instead, they treat the patient as a whole. In this clinical study, the therapeutic effects and partial or complete stabilization of the allergic condition of fourteen dogs divided into two groups were observed, compared, and evaluated.

## 1. Introduction

Food allergies in dogs are constantly increasing, and they are one of the most common causes of dermatological and gastrointestinal problems in small animals [1]. Food hypersensitivity is caused by an immunological reaction to an antigen present in the food [2]. The intestinal mucosa normally provides three functions. The first is to prevent the penetration of ingested antigens by an intact epithelial barrier, proper peristalsis, and mucosal glycocalyx. The second function is to promote the degradation of ingested antigens by gastric acid, pancreatic enzymes, and brush border enzymes. A third function is to facilitate the secretion of antigens from the mucosa into the intestinal lumen via IgA secretion [3]. During the allergic reaction, the immune system produces an antibody called immunoglobulin E (IgE) that binds to the surface of mast cells. If the exposure between the same antigen and the animal is repeated, the antigen binds to the antibody and causes degranulation of mast cells, followed by the release of inflammatory mediators causing inflammation. This “early” type I hypersensitivity reaction occurs only a few minutes or hours after an exposure of the body to an antigen. Food antigen hypersensitivity can gradually lead to other diseases, such as inflammatory bowel disease, etc. [3]. Clinical symptoms occur after antigen penetration through the intestinal wall and after its encounter with sensitized basophils or IgE bound to mast cells in the skin [1,3]

In dogs, beef, pork, chicken, and dairy products are considered the most common allergens [3,4]. Food hypersensitivity can occur at any age, from weaned puppies to old dogs whose feed composition was never changed. Approximately 30% of dogs with a confirmed diagnosis are younger than 1 year old [1]. Clinical signs include non-seasonal pruritus, which may respond positively or negatively to steroid therapy. Usually, this pruritic condition affects the ear, feet, inguinal region, axillary area, face, neck, and perineum. Then erythema, papules, alopecia, crusts, and hyperpigmentation may occur as a result of pruritus, and these symptoms are often accompanied by secondary bacterial or malassezia infection. In some patients, otitis externa is the only manifesting symptom [1,3]. Affected individuals may be prone to gastrointestinal symptoms, including vomiting, diarrhea, weight loss, abdominal pain, borborygm, frequent defecation, and flatulence that occur in 20–30% of affected patients [5,6,7,8]. Approximately 20–30% of dogs with food allergies also suffer from other allergies, such as hypersensitivity to flea bites or atopic dermatitis [4]. The long-term management and therapy require an individualized approach, which also includes special dog nutrition.

Treatment is usually composed of Janus kinase inhibitors (Apoquel tbl., Cytopint inj.), antibiotics, and corticosteroids, which are, unfortunately, associated with many side effects. Due to the risk of side effects from conventional Western medicine therapies and the request from some owners of animals that suffer from allergy for alternative therapies, a combination of acupuncture, phytotherapy, and nutrition was applied. The results of alternative therapy were compared and evaluated with the results of conventional therapy applied in allergic patients treated by conventional Western medicine therapy.

## 2. Materials and Methods

### 2.1. Dogs

In this research study, we included fourteen dogs divided into two groups of 7 individuals with confirmed food hypersensitivity and the same pruritic condition of the skin. In the first group, acupuncture and phytotherapy were applied as a treatment. The results were compared with the second observed group that received a conventional Western medicine therapy that included corticosteroids, antibiotics, and Janus kinase inhibitors.

### 2.2. Evaluation of Clinical Parameters

After receiving a detailed case history of each dog at clinical examination, hematological and biochemical analysis were performed. Blood collections were carried out via antecubital venipuncture after 12 h of fasting.

The blood sample was evaluated by ProCyte IDEXX Dx (IDEXX Laboratories, Westbrook, ME, USA), where counts of erythrocytes, reticulocytes, total leucocytes, platelets, hematocrit, hemoglobin concentration, and differential leukogram were assessed.

Biochemical analysis was performed from blood serum and evaluated by Catalyst IDEXX One (IDEXX Laboratories, USA), where the concentrations of liver enzymes (ALT, ALP), creatinine (CREA), urea (BUN), cholesterol (TC), pancreatic amylase (AMS) and lipase (LPS) were assessed in all observed dogs.

Cytological examination of skin lesions was performed in each dog by using an impression smear. The dermatological examination was composed of a microscopic assessment of skin scraping that confirmed or excluded the presence of parasites, and a skin swab culture that was performed for bacteriological and mycological laboratory analysis. Food hypersensitivity was confirmed in all patients by an ELISA blood test provided by the diagnostic laboratory LABOKLIN s.r.o., Bratislava, Slovakia. 

### 2.3. Diagnostics and Treatment Protocol

In the first group of patients, diagnostics were performed according to the principles of traditional Chinese medicine (TCM), consisting of Yin–Yang theory and five-element theory. The treatment was composed of acupuncture, phytotherapy and nutrition.

In the case of allergic skin manifestations and ear infections in dogs, there is usually an excess of Yang energy, which is characterized by damp and heat (inflammation, redness, and discharge), so in this case, animal food was characterized by cold types of food. For each dog, the diet was designed individually. In the summer, patients received neutral, cold, and moist food, so in this case, raw food was the most suitable. In the winter, the animals were fed warm and cooked food, which means that the dog’s food was selected according to its purpose in TCM theory (tonification, regulation, or purification of the body) and ensuring a balance between Yin and Yang energy.

Acupuncture was performed in this group of patients with disposable sterile acupuncture needles. Acupuncture point LI-11, which is ideal for swelling, fever, and also pruritus, was applied to all patients. Other acupuncture points, such as ST-36, LI-4, BL-23, and BAI HUI, were also applied to all patients. The other acupuncture points were selected for each individual according to its health condition and to eliminate the pathogenic factor, which, according to TCM, in case of allergy with subsequent dermatological problems, might be dampness, dryness, cold, heat, wind, or other harmful influences.

Another alternative therapeutic method applied to the first group of patients was phytotherapy. An individual combination of phytotherapy and acupuncture was used in all dogs in this group, where *Silybum marianum, Glycyrrhiza glabra, Curcuma longa*, and *Pleurotus ostreatus* were administrated to patients.

Diagnosis and treatment in the second group were performed in a standard manner. Antibiotics, corticosteroids, and Janus kinase inhibitors were applied as part of the therapy, according to the symptoms, type, and severity of the allergic manifestation in the animal. All dogs were fed a hypoallergenic diet.

### 2.4. Statistical Evaluation

Statistical evaluation of hematological and biochemical parameters before and after therapy was performed by paired t-test at a significance level of α = 0.05.

## 3. Results

All of the dogs that were included in the study are represented in Table 1.

### 3.1. The First Observed Group of Dogs

All animals suffered from allergic symptoms of a type 1 hypersensitivity reaction, manifested by dermatological symptoms, mainly as eye and ear discharge and pruritus. Dermatological lesions appeared mostly on the head, axillae, toes, and inguinal area. Pruritus with swelling of the skin around the eyes, ears, and papules occurred on the head. The skin showed signs of inflammation in the form of erythema, pustules, and scabs in these areas. Signs of erythematous and pruritic skin also appeared on the ventral abdomen, in the axillary area, or in the form of pododermatitis with intense finger licking. The clinical symptoms of all patients of the first group are summarized in Table 2, showing the individual symptoms in each patient before and after therapy. The table also shows the type of phytotherapy applied to each dog individually.

### 3.2. Clinicopathological Abnormalities

The hematological analysis of the first group of dogs is recorded in Table 3 where the largest changes occurred in the concentration of eosinophils, which were increased in six patients, and their average was 1.78 × 10^9^ /L compared to the standard, 0.1–1.49 × 10^9^/L. Three patients developed leukocytosis before treatment, which corresponded to a mean value of 17.1 × 10^9^/L compared to the norm of 5.05–16.76 × 10^9^/L. Neutrophilia occurred in two dogs, with a mean value of 13.33 × 10^9^/L compared to the standard 2.00–12.00 × 10^9^/L. After therapy, the levels of eosinophils (Eos), leukocytes (Leu), and neutrophils (Neu) were adjusted to the reference range.

The statistical analysis comparing individual hematological parameters before and after therapy revealed statistical significance for Eos, Leu, and Neu concentrations at a significance level of α = 0.05. The comparisons of erythrocyte (Ery) and lymphocyte (Lym) concentrations before and after therapy were not statistically significant.

The monitoring of biochemical parameters in the first group revealed increased serum concentrations of alanine aminotransferase (ALT) and alkaline phosphatase (ALP) in three dogs before therapy (Table 4). In the case of the liver enzyme ALT, the mean value (134.6 U/L) was increased compared to the standard deviation (ALT 10–125 U/L). The mean concentration of ALP (277.3 U/L) also represented an increase compared to the norm (ALP 23–212 U/L). The overall mean ALT (88.7 U/L) and ALP (190.4 U/L) levels of all patients were within the reference range. In all cases, elevated enzymes were adjusted to physiological levels after therapy. Pancreatic amylase (AMS) and lipase (LPS) were also increased in serum concentrations. The elevation of AMS occurred in three dogs. However, the average value of AMS (1244.2 U/L) was within the reference range (AMS 500–1500 U/L), and the average value of the three mentioned dogs was increased (1631 U/L). Elevation of pancreatic lipase (LPS) was observed in two patients, where their mean level (1928.5 U/L) was increased from normal (LPS 200–1800 U/L). After therapy, the serum levels of pancreatic enzymes were adjusted in all observed patients. In the case of urea (BUN) and creatinine (CREA), we recorded the elevation of urea only in one patient (32.2 mg/dL), which had no effect on the overall urea mean (18.8 mg/dL), where its reference range is 7–27 mg/dL. The creatinine (0.5–1.8 mg/dL) and cholesterol levels (110–320 mg/dL), in all cases, were within the reference range. The values of biochemical parameters in individual dogs before and after therapy are recorded in Table 4.

The statistical analysis of the comparison of individual biochemical parameters before and after therapy revealed statistical significance in the case of serum concentrations of ALP and LPS at the level of significance α = 0.05. The deviations in parameters such as BUN, CREA, ALT, AMS, and TC were statistically insignificant.

### 3.3. The Second Observed Group of Dogs

All animals from the second group suffered from allergy reactions of a first type of hypersensitivity. Dermatological symptoms appeared mainly on the head, axillae, inguinal area, and distal parts of the limbs, with intense pruritus, erythema, swelling, and pustules that were accompanied by secondary Malassezia infections. Dermatological symptoms before and after therapy for the second group are shown in Table 5. The table also captures individually applied conventional Western medicine drug therapy for individual dogs. Therapy was performed with corticosteroids, antibiotics, and, in some cases, Janus kinase inhibitors (Apoquel and Cytopoint).

### 3.4. Clinicopathological Abnormalities

Our evaluation of hematological analysis in dogs of the second group (Table 6) showed the most numerous changes in eosinophil concentrations, which were increased in five patients compared to the reference level (0.1–1.49 × 10^9^/L). Their average was 1.7 × 10^9^/L. In all cases, eosinophil levels were adjusted after therapy, but in two cases, the number of eosinophils still exceeded the average of the reference value (1.52 209 × 10^9^/L). In three dogs, the number of eosinophils decreased to the reference standard. In the case of leukocytes, only one patient (Patient 7) developed leukocytosis (17.24 × 10^9^/L) before treatment and also increased the number of neutrophils with a value of 13.54 × 10^9^/L compared to the norm (2.00–12.00 ×10^9^/L). After therapy, both leukocyte and neutrophil levels were adjusted to the reference range.

The statistical analysis of the comparison of individual hematological parameters before and after therapy revealed statistical significance in the second group of dogs in the case of Eos and Leu concentrations at the significance level α = 0.05. The comparisons of erythrocyte, neutrophil, and lymphocyte concentrations before and after therapy were not statistically significant.

Biochemical parameters in the second group of dogs (Table 7) revealed increased serum ALT concentrations in two patients, with a mean value of 136 U/L, compared to the normal value, which is 10–125 U/L. After therapy, ALT levels returned to the reference range (10–125 U/L). The levels of ALP were increased in four dogs before therapy, where their average was 312.25 U/L compared to the reference limit, 23–212 U/L. After treatment, the levels of ALP did not return to the reference range (23–212 U/L) and also increased in the other two dogs, thus exceeding the level of the reference standard. After therapy, the average value of ALP increased to 325.6 U/L. Because these animals were temporarily stabilized or still receiving ongoing therapy of corticosteroids, their persistent induction caused increasing ALP levels and other side effects. Incremental increases of pancreatic enzymes (AMS and LPS) were also observed. There was an increase in AMS levels in five patients before treatment, with the mean value of all dogs being 1628.4 U/L, which exceeded the reference standard of 500–1500 U/L. After therapy, the AMS value was stabilized within the reference standard in three dogs. In two dogs, its value increased even more, reaching an average value of up to 2237 U/L. The elevation of pancreatic lipase (LPS) was recorded in three patients, where their average level (1892 U/L) was increased compared to the norm (200–1800 U/L). After therapy, the elevated LPS levels improved in only one patient, but the LPS values increased in the other two dogs, so finally LPS values were increased in three animals, and their average value was 1986.6 U/L. An increase in the levels of urea and creatinine was also recorded in the level of urea and creatinine, where BUN concentration (2.5–9.6 mmol/L) was elevated before (9.9 mmol/L) and after (9.8 mmol/L) treatment in only one patient. Creatinine levels were altered before treatment in one patient (172 µmol/L) and after therapy in three patients, averaging an increased value of 178 µmol/L compared to the reference standard of 44–159 µmol/L. Cholesterol (TC) levels (110–320 mg/dL) were within the reference range in all cases.

The statistical analysis of the comparison of individual biochemical parameters before and after therapy revealed statistical significance in the case of serum concentrations of ALP and TC at the level of significance α = 0.05. The changes in parameters such as BUN, CREA, ALT, and AMS were statistically insignificant.

Figure 1 shows the percentage increase/decrease in the averages of hematological parameters in patients of both groups after treatment compared to the condition before therapy. Decreases in mean values occurred in both groups in leukocytes, neutrophils, and eosinophils after treatment. In contrast, an increase of lymphocytes was observed in both groups of dogs after treatment.

Figure 2 shows the percentage increase/decrease in the averages of biochemical indicators according to the analyzed groups after therapy compared to the condition before therapy. The most significant changes occurred with an increase of average values in the second group of dogs after therapy in the parameters ALP, LPS, TC, CREA, and BUN.

## 4. Discussion

There are many cases of allergies in dogs that can cause a hypersensitivity reaction to pollen, mites, or a particular food ingredient [1,3]. Typical symptoms of food allergy are itching and changes in the skin, which may be accompanied by symptoms in the gastrointestinal system [1]. As skin problems tend to be treated mainly symptomatologically, in most cases corticosteroids are the first and the most rapid therapeutic choice, which is unfortunately not always advantageous for the animal [9]. Janus kinase inhibitors are another therapeutic option used in conventional Western medicine that, in most cases, stops pruritus but does not solve the problem. In addition to the characteristic clinical findings in dogs with hypersensitivity reactions, there are often corresponding findings of hematological and biochemical parameters. Many authors using blood tests in allergic patients describe the occurrence of eosinophilia [10,11,12].

Our study also identified increased eosinophil levels at the beginning of treatment in dogs from all groups. During therapy, eosinophil concentrations decreased in all dogs, with the exception of two dogs in the second group, which was treated with glucocorticoids; for those two dogs, eosinophil levels still exceeded the reference standard. Statistically significant results were also observed in other hematological parameters, especially in the levels of leukocytes and neutrophils, and this was consistent with other studies [13,14]. In all observed cases, the abovementioned parameters were improved by the conventional or alternative medicine therapy. Similarly, many authors report that, in the case of allergic dogs, they have observed a decrease in the initial elevated concentrations of hematological parameters during long-term treatment [9,15]. An adjustment in both groups for lymphocyte concentrations, which were reduced in some patients at the beginning of treatment, was also observed in our research. Although lymphocyte levels are not considered as a characteristic indicator of allergy in hypersensitive patients, some authors have reported lymphopenia associated with presumed immunosuppression, which may be related to the disease [16]. When comparing the results of hematological parameters, we can conclude that the adjustment of the indicated concentrations in dogs occurred in both groups, thus confirming that alternative methods of therapy obviously have a desirable effect in the treatment of allergic symptoms. In biochemical parameters, we recorded several parameter alterations in all patients at the beginning of therapy, mainly in ALP, ALT, AMS, and LPS. Their serum concentrations were increased to varying degrees in each group at the beginning of therapy. The largest differences before and after therapy were observed in individuals in ALP levels. While elevated ALP levels decreased after treatment in the first group, in the second group of dogs that were treated also with glucocorticoids, its serum concentration increased significantly. Other authors have reported an increase in ALP after various forms of drug therapy, especially after the administration of glucocorticoids [17]. The well-known side effects of glucocorticoid therapy can be observed in all organ systems of patients, because their long-term administration damages the bones and endocrine glands, affects hematopoiesis, interferes with liver metabolism, and impairs pancreatic function [9]. The consequences of glucocorticoid therapy are also marked by an excess of pancreatic enzymes (AMS and LPS), which also occurred in patients from the second study group after long-term administration of prednisone. In some dogs of both study groups, some of these pancreatic enzymes were elevated before therapy. While AMS and LPS levels of the first study group adjusted to their normal values after therapy, the second group of dogs had a tendency to increase their serum concentrations in some patients. These results show that TCM and phytotherapy have a lower risk of side effects in the treatment of allergy [18].

Nutrition composed of balanced feeding in dogs with hypersensitivity has been an important part of treatment. An appropriate diet was included in both groups, where the elimination diet was ordered [19]. The first step in treatment for food allergies is to remove an inappropriate feed [20]. The correct diet was followed in both groups. In the first group, feeding was adjusted according to TCM principles. In the second group, we followed the recommended elimination diet, and the animals were fed by hypoallergenic dog food. However, the patient’s diet may change over time according to his or her current health condition and may also include raw food ingredients. If biologically suitable raw foods (BARFs) are selected, the diet should be sufficiently balanced, mixed with fresh and safe ingredients, and supplemented with nutritional supplements [21]. According to studies, BARF raw food is most suitable for dogs because they have a digestive system developed to receive raw food. With this change in diet, we can better manage many chronic digestive problems. According to TCM, food variability is important to balance energy and ensure animal comfort and health. In China, foods that have been used for the treatment of diseases are classified according to their energy properties [22]. Due to the fact that TCM treats each individual independently, the most important is the patient’s current energy and health status during the examination [23]. We also adapted the food to the individual needs of our patients in the first group.

One of our applied alternative methods in the first group of patients was TCM, where we used a combination of acupuncture and phytotherapy. There are many studies that examine the effectiveness of acupuncture in allergies. The antipruritic effects of acupuncture have been demonstrated in artificial pruritus in healthy placebo-controlled studies [24]. Acupuncture has shown better results than antihistamines and preventive antipruritic systemic drugs [24,25]. A pilot study showed that acupuncture reduced basophil activation in allergic patients [26]. Another study showed that acupuncture reduced serum IgE levels, as well as mRNA expression of the pro-inflammatory cytokines IL-4, IL-8, and TNF-α [27]. Acupuncture has modulatory effects on cytokines, hormones, neurotransmitters, growth factors, and oxidative factors [28]. The mechanism of acupuncture’s antipruritic effect in the brain has been observed by using functional magnetic resonance imaging [24]. Researchers have shown a reduction in pruritus-induced activation in the putamen, insula, premotor, and prefrontal cortex, as well as an itching reduction [29]. Acupuncture treatment can be effective in modulating central sensitivity and is effective in reducing stress and treating emotional disorders. It is also used to treat anxiety and depression [30]. Acupuncture affects the limbic system, which is responsible for stress-related symptoms, so stress may be responsible for the recurrence of allergies and pruritus in the patient [31]. Several studies have examined the specific acupuncture point LI-11 (Qū Shī) in pruritic patients. They applied it as a basic antipruritic point, whereby stimulating this point with an acupuncture needle, pressure, or moxa, they reduced itching in patients [27,32]. Point LI-11 is an important point in the colon’s pathway that can cleanse heat (Yǎng) and is used for dry, pruritic, scaly, and inflamed skin. This point is also able to reduce mRNA expression of pro-inflammatory cytokines and proteins [24]. According to these problems, acupuncture, which improves the symptoms of atopic dermatitis and prolongs the duration of action without serious side effects, could be effective for patients. As part of phytotherapy, we applied beta-glucan, turmeric, silymarin, and licorice. *Glycyrrhiza glabra* contains plant constituents that inhibit renal 11β-hydroxysteroid dehydrogenase activity, thereby reducing the conversion of cortisol to cortisone and resulting in high renal levels of cortisol, which is available for binding to mineralocorticoid receptors. As it has the effects of mineralocorticoids and glucocorticoids, it increases the duration and effectiveness of cortisone in the body. It reduces the secretion of gastric juices and has antiulcer effects. In this study, it was applied to three animals because it provides spleen and Qi energy tonification, moisturizes the lungs, and stops coughing; additionally, it is suitable for both asthmatic and pruritic conditions. According to TCM, it eliminates toxic heat, which is one of the pathogenic factors. It has an anticonvulsant effect and has harmonizing effects on the body. However, too high of a dose of licorice (*Glycyrrhiza glabra*) can lead to edema, although deglycyrrhizinated versions of licorice are also available. Licorice contains potassium levels that may cause an imbalance between sodium and potassium, leading to cardiac arrhythmias and hypertension. It is also contraindicated in type I diabetes due to its sweet taste [33]. Therefore, its use and dosage should be considered, especially in geriatric patients and in animals with heart and kidney disease, so patients should be monitored regularly [34]. The combination of licorice and beta-glucan in one preparation was applied to the dogs. This product was administered to patients for two weeks. If an animal was without side effects, we continued to administer this phytotherapeutic combination. If the animal had side effects, we changed the product for another one containing beta-glucan and vitamin C. We administered the products in the first observed group of patients for 3 months. Beta-glucans are immunomodulators and activators of white blood cells, especially macrophages, which can significantly affect the body’s regeneration process [35]. They have immunomodulatory effects, stimulate the immune system, increase the body’s resistance, and protect the body from infection [36,37]. We applied it to two animals in combination with acupuncture therapy. Its practical use generally alleviates clinical symptoms of diseases, accelerates the healing process, reduces secondary infection, and increases the effectiveness of causal therapy and prognosis of healing. Beta glucan also has an adjuvant effect in increasing the effectiveness of treatment with other drugs, such as antibiotics and natural supplements [38]. It is also safe and non-toxic for animals [39]. Based on these positive effects, we applied beta-glucan as a complementary therapy in the first group of dogs. Silymarin was chosen for three patients in the first group because it has a detoxifying, calming, and antipruritic effect [40]. It is a typical remedy for liver disease. According to TCM, the liver belongs to the wood element. The characteristic emotional expression of this element is aggression and anger. According to TCM, pruritus is bound to the liver, so if there is a severe itchy condition, it is good to detoxify the liver and gradually eliminate the accumulated “Wind” (Fēng) from the liver, which is one of the pathogenic factors affecting the patient’s health [41]. Turmeric (*Curcuma longa*) was administered to two patients in the first group of animals suffering from food allergy. It is traditionally used to treat pain, fever, allergic, and inflammatory diseases such as bronchitis, arthritis, and dermatitis. Its active ingredient, curcumin, is particularly effective in alleviating immune disorders, including allergies. According to studies, turmeric significantly alleviates the symptoms of food allergy and also inhibits IgE and IgG1 levels, and this confirms the significant alleviation of food allergy symptoms also recorded in a mouse model. Turmeric as an antiallergic agent has shown an immunoregulatory effect by maintaining an immune balance between Th1 and Th2. Its administration is useful in alleviating Th2-mediated allergic disorders, such as food allergy, atopic dermatitis, and asthma [42].

Conventional Western medicine drug therapy applied to the second group of dogs was able to stabilize the patient’s condition, but only temporarily, so recurrence of the allergic symptoms developed. Some dogs of this study group are still in the ongoing therapy stage, so the treatment is still presented. In this group, 40% of individuals experienced drug side effects such as obesity, joint problems, slowed wound healing process, and deterioration in the biochemical parameters of ALP and TC. Many authors in their research also demonstrate the occurrence of these side effects and also describe other individual symptoms that are typical of long-term glucocorticoid therapy [16].

## 5. Conclusions

The observations made in this study cannot be presented as unequivocal and comprehensive instruction in solving the problem of allergy in dogs. The aim of this study was to show and pay attention to the existence of alternative diagnostic and therapeutic treatment methods that have been used in the past and which are becoming nowadays very popular nowadays. According to much worldwide research on traditional therapeutic methods, such as acupuncture, phytotherapy and others, which have been used for centuries, such practices are also very beneficial in the treatment of allergies. If these non-invasive methods are used properly, they can significantly improve or even stabilize many health problems and diseases with minimal side effects. Alternative options are relatively safe, without the risk of side effects, and they do not harm the body. They can be used separately or in combination with other therapies; and they can be used as an independent or complementary therapy for both animals and humans.

## Figures and Tables

**Figure 1 animals-12-01832-f001:**
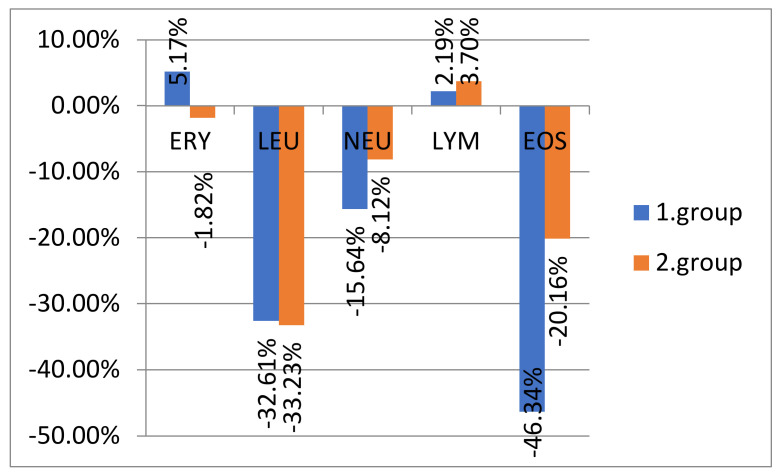
Percentage increase/decrease in averages of hematological indicators according to the analyzed groups after therapy. Legend: Ery, erythrocytes; Leu, leucocytes; Neu, neutrophils; Lym, lymphocytes; Eos, eosinophils; “1. group” is the group of dogs treated with acupuncture, phytotherapy and nutrition; “2. group” is the group of dogs treated with conventional Western medicine drug therapy.

**Figure 2 animals-12-01832-f002:**
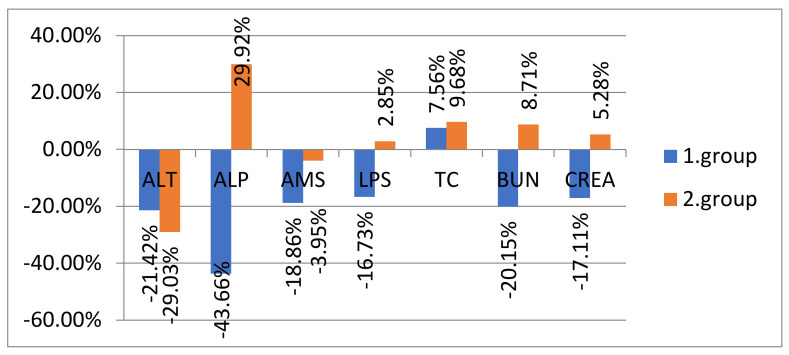
Percentage increase/decrease in averages of biochemical indicators in both groups after therapy. Legend: ALT, alanine aminotransferase; ALP, alkaline phosphatase; AMS, amylase; LPS, lipase; TC, total cholesterol; BUN, urea; CREA, creatinine; “1. group” is the group of dogs treated with acupuncture, phytotherapy and nutrition; “2. group” is the group of dogs treated with conventional Western medicine drug therapy.

**Table 1 animals-12-01832-t001:** The first and the second group of dogs included in the study.

	Group 1			Group 2		
	Breed	Age (Years)	Gender	Breed	Age (Years)	Gender
**1**	Dogue de Bordeaux	5	F	Crossbreed dog	5	F
**2**	Chihuahua	2	F	Magyar Vizsla	5	M
**3**	Boxer	8	F	French Bulldog	5	M
**4**	German Shepherd	9	M	Pug	3	F
**5**	Maltese dog	5	M	Labrador retriever	10	M
**6**	Yorkshire Terrier	6	F	Schnauzer dog	8	M
**7**	Crossbreed dog	9	M	Crossbreed dog	8	F

**Table 2 animals-12-01832-t002:** Clinical symptoms before therapy, and condition after application of acupuncture and phytotherapy in the first observed group.

Dog	Clinical Symptoms before Therapy	Phytotherapy	Condition after Therapy
**1**	Erythema and alopecia of the abdomen and limbs, pruritus, and nasal discharge	Silybum marianum	Without clinical symptoms
**2**	Erythema and alopecia of the abdomen and pruritus	Silybum marianum	Without clinical symptoms
**3**	Erythema in the axillae and pruritus	Curcuma longa	Alleviated pruritic condition
**4**	Erythema in the axillae and abdomen and pododermatitis	Pleurotus ostreatus	Without erythema and alleviated pododermatitis
**5**	Pruritus of mouth, axillae, abdomen, and limbs	Glycyrrhiza glabra	Alleviated pruritic condition
**6**	Pruritus around eyes and ears and nasal discharge	Silybum marianum	Without clinical symptoms
**7**	Pruritus of mouth and pododermatitis	Pleurotus ostreatus	Without clinical symptoms

**Table 3 animals-12-01832-t003:** Hematological examination before (**B**) and after (**A**) therapy in the first group of patients.

Dog	Ery 5.65–8.87 × 10^12^/L	Leu 5.05–16.76 × 10^9^/L	Neu 2.00–12.00 × 10^9^/L	Lym 0.50–4.90 × 10^9^/L	Eos 0.10–1.49 × 10^9^/L
	B/A	B/A	B/A	B/A	B/A
**1**	5.73/5.78	15.46/10.72	9.55/8.03	2.65/2.12	*** 1.92**/1.35
**2**	6.82/7.09	*** 16.93**/10.22	10.12/7.98	3.34/3.02	*** 1.65**/0.75
**3**	7.38/7.78	14.45/9.18	5.00/3.98	4.20/4.34	*** 1.55**/0.81
**4**	5.35/5.68	*** 17.39**/11.39	*** 14.15**/12.00	0.52/0.70	*** 1.77**/0.89
**5**	7.56/8.14	13.78/11.29	10.56/9.03	2.56/2.83	*** 1.89**/1.11
**6**	5.23/5.66	10.59/8.00	8.00/6.73	2.35/2.82	1.29/0.57
**7**	5.65/5.85	*** 16.98**/10.35	*** 12.52**/11.22	0.50/0.52	*** 1.95**/0.97

Legend: Ery, erythrocytes; Leu, leucocytes; Neu, neutrophils; Lym, lymphocytes; Eos, eosinophils; trillions (10^12^)/liter; billions (10^9^)/liter. Values out of reference range are marked with an asterisk *****. Bold: Biochemical parameter out of reference range.

**Table 4 animals-12-01832-t004:** Biochemical examination before (**B**) and after (**A**) therapy in the first group of patients.

Dog	ALT 10–125 U/L	ALP 23–212 U/L	AMS 500–1500 U/L	LPS 200–1800 U/L	TC 110–320 mg/dL	BUN 7–27 mg/dL	CREA 0.5–1.8 mg/dL
	B/A	B/A	B/A	B/A	B/A	B/A	B/A
**1**	13/20	76/82	822/850	1713/1618	121/119	12/10	1.5/1.2
**2**	45/41	50/64	*** 1520**/980	205/201	41/44	8.2/7.9	1.3/1.1
**3**	81/85	*** 254**/155	692/690	1560/1553	210/240	*** 32.2**/26.1	0.99/0.62
**4**	*** 144**/98	*** 352**/191	*** 1700**/1190	*** 1905**/1450	256/283	15.6/13.4	0.8/0.7
**5**	78/76	210/98	1490/1325	1470/1426	265/268	14/12	1.6/1.5
**6**	*** 128**/79	165/72	812/865	1786/1245	196/194	21/20	0.7/0.5
**7**	*** 132**/89	*** 226**/89	*** 1674**/1167	*** 1952**/1337	220/260	26/16	1.7/1.5

Legend: ALT, alanine aminotransferase; ALP, alkaline phosphatase; AMS, amylase; LPS, lipase; TC, total cholesterol; BUN, urea; CREA, creatinine; U/L, units/liter; mg/dL, milligrams/ deciliter. Values out of reference range are marked with an asterisk *. Bold: Biochemical parameter out of reference range.

**Table 5 animals-12-01832-t005:** Clinical symptoms before therapy and after treatment applied to the second group of dogs.

Dog	Clinical Symptoms before Therapy	Therapy	Status after Therapy
**1**	Otitis externa, pododermatitis	ATB and Prednisolone	Without clinical symptoms
**2**	Pruritus of abdomen	Cytopoint and Prednisolone	Alleviated pruritic condition
**3**	Pruritus of eyes, mouth and anus, otitis externa	Apoquel, ATB, and Prednisolone	Ongoing therapy
**4**	Pododermatitis, pruritus of mouth and anus	Cytopoint, ATB, Prednisolone	Ongoing therapy
**5**	Otitis externa, pododermatitis, pruritus of abdomen, axillae, and limbs	Apoquel, ATB, and Prednisolone	Ongoing therapy
**6**	Pruritus of mouth and abdomen	Apoquel, ATB, and Prednisolone	Ongoing therapy
**7**	Pruritus of mouth, eyes, nasal discharge	ATB and Prednisolone	Without clinical symptoms

**Table 6 animals-12-01832-t006:** Hematological analysis before (B) and after (A) therapy in the second group of patients.

	Ery 5.65–8.87 ×10^12^/L	Leu 5.05–16.76 ×10^9^/L	Neu 2.00–12.00 ×10^9^/L	Lym 0.50–4.90 ×10^9^/L	Eos 0.10–1.49 ×10^9^/L
Dog	B/A	B/A	B/A	B/A	B/A
**1**	5.69/5.72	8.05/6.02	3.06/2.66	1.30/1.35	*** 1.58**/1.47
**2**	6.25/6.41	9.25/6.12	7.36/6.93	2.35/2.14	*** 1.74**/1.21
**3**	6.48/6.7	12.48/7.5	6.42/6.18	4.32/3.92	1.03/0.98
**4**	7.35/7.40	10.56/9.2	6.0/5.7	3.56/5.13	*** 1.95**/**1.54 ***
**5**	5.73/5.61	13.24/8.3	10.94/10.13	1.39/1.26	0.34/0.25
**6**	5.65/5.64	16.54/9.16	11.03/10.76	1.52/1.02	*** 1.67**/0.93
**7**	6.21/5.79	*** 17.24**/16.12	*** 13.54**/11.25	0.52/0.59	*** 1.56**/**1.50 ***

Legend: Ery, e Legend: Ery, erythrocytes; Leu, leucocytes; Neu, neutrophils; Lym, lymphocytes; Eos, eosinophils; trillions (10^12^)/liter; billions (10^9^)/liter. Values out of reference range are marked with an asterisk *****. Bold: Biochemical parameter out of reference range.

**Table 7 animals-12-01832-t007:** Values of the biochemical parameters in the second group of dogs before (**B**) and after (**A**) therapy.

Dog	ALT 10–125 U/L	ALP 23–212 U/L	AMS 500–1500 U/L	LPS 200–1800 U/L	TC 110–320 mg/dL	BUN 2.5–9.6 mmol/L	CREA 44–159 µmol/L
	B/A	B/A	B/A	B/A	B/A	B/A	B/A
**1**	99/38	114/**222** *****	*** 1540**/1322	*** 1850**/**1970*******	258/275	8.6/**9.8** *****	56/58
**2**	85/79	32/52	*** 1600**/1420	*** 1850**/1775	155/158	2.5/3.8	89/81
**3**	52/56	*** 286**/**282*******	*** 1746**/**1658*******	462/493	262/271	6.9/9.1	144/**165** *****
**4**	33/41	*** 258**/**293*******	1418/1324	843/865	158/190	8.9/8.6	132/129
**5**	55/93	149/**305** *****	934/955	1266/1178	255/283	*** 9.9**/9.1	*** 172**/135
**6**	*** 131**/98	*** 252**/**394*******	*** 2336**/**2816*******	*** 1976**/**2038*******	253/263	2.4/2.1	155/**183** *****
**7**	*** 141**/18	*** 453**/**458*******	*** 1825**/1454	1739/**1952** *****	239/293	5.6/6.2	142/**186** *****

Legend: ALT, alanine aminotransferase; ALP, alkaline phosphatase; AMS, amylase; LPS, lipase; TC, total cholesterol; BUN, urea; CREA, creatinine; U/L, unit/liter; mg/dL, milligram/deciliter; mmol/L, millimole/liter; µmol/L, micromole/liter. Values out of reference range are marked with an asterisk *****. Bold: Biochemical parameter out of reference range.

## Data Availability

All data are provided in the article.

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
