# Peer review of "Comparative Study of Classical and Alternative Therapy in Dogs with Allergies"

_animals, 2022, doi:10.3390/ani12141832_

Round 1
Reviewer 1 Report
I strongly recommend to improve english language editing. There are some specific detail to be corrected:
Line 20: Janus kinase inhibitors
Line 73: research study
Line 121 and Table 2: Silybum marianum, Glycyrrhiza glabra
Line 151 choose conventional acronyms for Ery, Leu, Meu, Lym and Eoz
Line 173 urea diameter?
Line 340 Qū Shī? 祛湿
Line 340 Yǎng 痒
Line 348 plant names not need capital letters, are common sustantives
Line 352 Qì 气
Line 381 "Wind" Fēng 风
I can not judge the scientific bases of phytotherapy and acupuncture. However the authors compare 2 groups of patients with food allergy treated with 2 different approaches, and provide their results and interpretation. Scientific evidence is very limited by means of the study design:
- Number of patients
- Inclusion and exclusion criteria
- Random distribution
- Double-blind
However, due to the interest of the subject it does not disqualify to be published.
Reviewer 2 Report
Manuscript Review for Animals-1739603
Overall Summary of Review
This is a very small but very interesting and potentially valuable clinical research study in dogs suffering food-based allergies and comparing traditional Chinese medicine combined with phytotherapy and nutrition to conventional Western medicine drug therapies for allergies. Although the study is small (7 dogs from unique breeds in each group) and therefore very surely must have high inherent biological variability in the biomarkers and other physiological measurements simply due to dog breed differences, this is still a very well-developed study of two active comparators (conventional Western drug therapy versus a traditional Chinese medicine approach with acupuncture, phytotherapy and nutrition). Any future study would be greatly strengthened by incorporating the STRICTA acupuncture guidelines; using the same purebred breed in each group; and choosing one type of phytotherapy rather than 3 or 4. The fact that of 7 dogs, only a few each received a unique phytotherapy, and since these phytotherapies are very unique in their effects on biochemistry and physiology, the ability to detect statistically meaningful differences for each phytotherapy treatment is even more reduced. Because of the variability, development of a more strict and well-defined protocol would also greatly improve any future study, with one well defined set of acupuncture treatments and one well-defined type of phytotherapy (or a combination of phytotherapeutics but always the same dose and type of phytotherapy), but even so, this is a reasonable example of what is now referred to as a small “pragmatic clinical trial” in a single university-based veterinary clinic. It is very clear that the authors have a strong command of the science and clinical aspects (diagnosis, treatment) of the study and these are very well described, and even though the study is small, it does provide good initial data that can be used for power analysis for future larger pragmatic clinical outcome active comparator studies. The study would have greatly benefited from using a single well-described alternative medicine protocol, since there are literally hundreds of alternative medicine treatments, including physical medicine, botanical medicine, nutritional medicine, acupuncture and east Asian medicine treatments, other traditional medicines like Ayurvedic medicine, Etc. That is why in future it would be better to define the protocol and apply it to a dozen dogs with allergies to observe the effects.
This article must be edited by a professional editing service as this is clearly written by non-native English speakers and is filled with literally hundreds of spelling, syntax and grammatical errors and this is totally unacceptable, making a huge amount of work for the reviewer addressing all these basic language errors. This article MUST be gone through literally “line by line” and corrected for spelling, syntax and grammar errors. The reviewer has provided many examples of spelling, syntax and grammar errors but cannot possibly correct all of them.
Detailed Remarks:
The authors use the term” classical” medicine where they should be using the term “conventional” medicine, please replace “classical” medicine with “conventional” medicine
Line 9: Sentence needs a period at the end.
Line 11: Consider replacing “Classical medicine can treat symptoms but cannot solve the problem” with “Conventional Western medicine can treat symptoms but often does not identify and resolve the underlying problem.”
Line 21: Replace “This comparative clinical study focused on the” with “This comparative clinical outcomes study focused on the”
Line 22: Replace: “where combination” with “where a combination”
Line 24-26: Please rewrite these lines: “In this clinical study were observed, compared and evaluated the therapeutic effects, and partial or complete stabilization of the allergic condition of fourteen dogs divided into two groups.” with “In this clinical study the therapeutic effects and partial or complete stabilization of the allergic condition of fourteen dogs divided 25 into two groups were observed, compared and evaluated.
Line 27: Replace “Traditional Chinese medicine” with “traditional Chinese medicine”
Line 32: Replace “causes of dermatological, and gastrointestinal problems” with “causes of dermatological and gastrointestinal problems”
Line 33: Replace “received by the feed” with “present in the food”
Line 35: “Replace “by intact epithelial barrier” with “by an intact epithelial barrier”
Line 38: Replace “lumen by IgA secretion” with “lumen via IgA secretion”
Line 39: Replace “the immune system, produces antibody” with “the immune system produces an antibody”
Line 40-42: Replace “If the exposition between the same antigen and the animal is repeated, antigen bindes to antibody that cause degranulation of mast cells and following release of inflammatory mediators causing inflammation” with “If the exposure between the same antigen and the animal is repeated, the antigen binds to the antibody and causes degranulation of mast cells followed by release of inflammatory mediators causing inflammation”
Line 45: Replace “Clinical symptoms occure” with “Clinical symptoms occur”
Line 42-45: Replace “This “early“ type I of hypersensitive reaction is formated only few minutes or hours after an exposure of the body to an antigen.” With “This “early“ type I hypersensitivity reaction occurs only a few minutes or hours after an exposure of the body to an antigen.”
Line 60-61: Replace “what means that allergic diseases have been serious disabilities that affected individuals suffer 61 for the rest of their lives.” With “which means that allergic diseases result in serious disabilities that affected individuals suffer for the rest of their lives.”
Line 62-63: Replace “The long-term management and therapy requires an individual approach, which also includes special dog nutrition.” With “The long-term management and therapy requires an individualized approach, which also includes special dog nutrition.”
Line 65: Replace “antibiotics and corticoids” with “antibiotics and corticosteroids”
Line 66-70: Replace “Due to risk of side effects of conventional therapy and the request of some owners of animals that suffer from allergy was alternative type of therapy provided, where combination of acupuncture, phytotherapy and nutrition was applied. Results of alternative therapy were compared and evaluated with the results of conventional therapy applied in allergic patients treated by classical medicine.” with “Due to the risk of side effects from conventional medicine therapies and the request from some owners of animals that suffer from allergy for alternative therapies, a combination of acupuncture, phytotherapy and nutrition was applied. Results of alternative therapy were compared and evaluated with the results of conventional therapy applied in allergic patients treated by conventional medicine.”
Line 79-80: Table 1 should be placed in the Results section and should not appear in the Materials and Methods section. Please move accordingly with associated text.
Line 79: Replace “The first and the second group of dogs included to study research.” With “The first and the second group of dogs included in the study.”
Line 82-83: This sentence make no sense in English and must be rewritten. Please consider changing “Blood collection was carried out from vena cephalica antebrachii after a previous 12-hour hunger strike.” to “Blood collections were carried out via antecubital veinipuncture after 12 hours of fasting.”
Line 86-87: Replace “haematocrit, haemoglobin concentration and differential leukogram were assessed.” with “hematocrit, hemoglobin concentration and differential leukogram were assessed.”
Line 97: Under Diagnostics and Treatment Protocol please describe the background and qualifications of the veterinarian administering acupuncture and other traditional Chinese medicine modalities. What degrees do they have in Acupuncture and East Asian Medicine and what are their certifications in what form of traditional Chinese medicine.
Line 99: Replace “Traditional Chinese medicine TCM consisted of Yin-Yang theory and five element theory.” With “traditional Chinese medicine TCM consisted of Yin-Yang theory and five element theory.” Traditional is not capitalized unless it is the beginning of a sentence.
Line 105-107: Replace “In this case was animal feed by cold types of food. For each patient was a diet approached individually and the feed ration was also adjusted to the season and energy status of the animal.” With “In this case animal food was characterized by cold types of food. For each dog the diet was designed individually and the food ration was also adjusted to the season and energy status of the animal.”
Line 107-100: Replace “In the summer, patients received neutral, cold and moist food so in this case the raw food was the most suitable. In winter, the animals were fed by warm and cooked food, what means that the patient's feed was selected according to its purpose (tonification, regulation, or purification of the body) and ensuring a balance between Yin and Yang 110 energy.” With “In the summer, patients received neutral, cold and moist food, so in this case raw food was the most suitable. In winter, the animals were fed warm and cooked food, which means that the dog’s food was selected according to its purpose according to its purpose in TCM theory (tonification, regulation, or purification of the body) and ensuring a balance between Yin and Yang energy.”
Line 115-118: Replace “The other acupuncture points were selected for each individual according to its health condition, so elimination of the pathogenic factor, which according to the TCM in case of allergy and subsequent dermatological problem might be Damp, Dryness, Cold, Heat, Wind or noxa was performed.” with “The other acupuncture points were selected for each individual according to its health condition and to eliminate the pathogenic factor, which, according to the TCM in case of allergy with subsequent dermatological problems, might be Dampness, Dryness, Cold, Heat, Wind or other harmful influences.”
Line 140-141: Replace “The table also shows a type of phytotherapy applied in each patient individually.” with “The table also shows the type of phytotherapy applied to each dog individually.
Table 2: replace “Glychirhiza glabra” with “Glycyrrhiza glabra”
Line 151-152: Please spell out the full name for eosinophils, Leukocytes and Neutrophils in the main body of the Results and do not use these abbreviations. Those who do not know these abbreviations will automatically become confused. The goal is to make everything as clear and simple as possible: “After therapy Eoz, Leu and Neu levels were adjusted to the reference range.”
Line 158-176: The first time an abbreviation is used you must first spell out the full name of the enzyme or biomarker and then you can use the abbreviated version from then on. Example, when alanine aminotransferase first appears say “alanine aminotransferase (ALT),” and from then on you can use the abbreviation.
Table 3 must have a legend at the bottom describing what the scientific numbers represent, what the units are (K/uL; /L) and also describing the full name for the various cell types and then there abbreviation: Leu: Leukocyte; Ery: Erythrocyte, etc. you must also indicate which comparisons were significant and at what level (alpha = 0.05) and place an Asterisk next each before/after comparison that was significant.
Table 4: must also have a legend that includes the full names for the biomarkers (ALT: alanine aminotransferase (ALT); ALP: alkaline phosphatase (ALP); etc. you must also indicate which comparisons were significant and at what level (alpha = 0.05) and place an Asterisk next each before/after comparison that was significant. Also, you do not show the overall mean and standard deviation for these 7 dogs for each biomarker tested. Please consider having another column that has the overall mean, the standard deviation, and the p value listed for each test that was performed. It is not clear how to structure this in the existing column or this can just be stated in the Results text along with the mean and standard deviations and p values, as you have done for the overall means.
Line 142: Replace “after application” with “after application”
Line 173: Replace “which had no effect on the overall urea diameter (18,8 mg/dL” with “which had no effect on the overall urea parameter (18,8 mg/dL)”
Line 183-189: Replace “All animals of the second group suffered from allergy reaction of a first type of hypersensitivity, Dermatological symptoms appeared mainly on the head, axillae, inguinal and distal parts of the limbs, and following intense pruritus, erythema, swelling and pustules accompanied by secondary Malassezia infection were presented. Dermatological symptoms before and after therapy of the second group shows table 5, The table also captures individually applied therapy for individual patients. Therapy was performed with corticoids, antibiotics and in some cases Janus kinase inhibitors (Apoquel, Cytopoint).” with “All animals from the second group suffered from allergy reactions of a first type of hypersensitivity. Dermatological symptoms appeared mainly on the head, axillae, inguinal area and distal parts of the limbs, with intense pruritus, erythema, swelling and pustules accompanied by secondary Malassezia infections. Dermatological symptoms before and after therapy for the second group are shown in Table 5. The table also captures individually applied conventional Western medicine drug therapies for individual dogs. Therapy was performed with corticosteroids, antibiotics and in some cases Janus kinase inhibitors (Apoquel, Cytopoint).
Line 191: Replace “Clinical symptoms before therapy and treatment applied in the second group of patients.” with “Clinical symptoms before therapy and after treatment applied to the second group of dogs.
Table 6 again perhaps having a bar separating the individual before/after values for each cell type examined and under the bar listing the overall means for before and after, the overall standard deviations for before and after, and the p values is needed for overall comparison of means and standard deviations. Also, there must be a legend with ALL tables that explains any abbreviations use (i.e., Eoz; Eosinophils; Lymphocytes, Lym; etc.) and explanation of what the numerical values (/L and K/uL) represent exactly.
In Graph 1 and Graph 2, please make sure that in the legend you include a description of what Group 1 and Group 2 represent so that an individual can just look at the graph and know that Group 1 is acupuncture with phytotherapy and nutrition and Group 2 is conventional Western medicine drug therapy. Also, in the legend make sure to list the full names of all the abbreviations so that an individual reading the paper can look just at the Graph or Table and understand everything without having to go looking through the text.
In the Results section please do not include any discussion of the individual subgroups of animals and their increases or decreases. Just report the mean, standard deviation and statistical significance for each group before and after therapy. Discussing subgroups within each treatment group only leads to more confusion when reading this paper. One must acknowledge this is a VERY small clinical outcome interventional study with a very small n value, so one must not try to draw too many conclusions, especially since the biological and genetic variation among these dog breeds is already very different, as is their immune status and physiology. The baseline values for the 7 dogs in each group demonstrate that they have wide variation in most of these parameters that does not involve the treatment interventions, and inter-animal variability will be the largest effect leading to non-significance of findings to too such a small n size.
Same as above for Table 7 and all other Table; you must show the overall means, the overall standard deviations, and the p values associated for eachbefore/after comparison.
Line 240-242: Please make sure in the Table legend to fully define the shortened terms for ER, LEU, Neu, LYM, EOS: The abbreviation and then the full name should be in the legend.
Line 260-261: Replace “which may be accompanied by symptoms of the gastrointestinal system” with “which may be accompanied by symptoms in the gastrointestinal system
Line 262: Replace “corticoids” with “corticosteroids”
Line 263: Replace “not always advantageous for an animal” with “not always advantageous for the animal”
Line 264-265: Replace “which in most cases stops pruritus but does not solve a problem.” with “which in most cases stops pruritus but does not solve the problem.”
Line 267-268: “Many authors from blood tests in allergic patients describe the occurrence of eosinophilia” with “Many authors using blood tests in allergic patients describe the occurrence of eosinophilia”
Line 269-272: Replace “In our study research was also observed increased eosinophil level at the beginning of treatment in patients of all groups. During therapy, eosinophil concentrations decreased in all patients, but in 2 dogs in the second group, treated with glucocorticoids, stayed their levels still exceeded from the reference standard.” with “In our study also identified increased eosinophil levels at the beginning of treatment in dogs from all groups. During therapy, eosinophil concentrations de creased in all dogs, but in 2 dogs in the second group treated with glucocorticoids, eosinophil levels still exceeded the reference standard.
Line 272-274: Replace “Also were recorded statistically significant results in other hematological parameters, especially in the levels of leukocytes and neutrophils, which also confirmed other authors in their work” with “statistically significant results were also observed in other hematological parameters, especially in the levels of leukocytes and neutrophils, which was consistent with other studies”
Line 274-275: Replace “In all observed cases, the mentioned parameters were adjusted by therapy.” With “In all observed cases, the above-mentioned parameters were improved by the conventional or alternative medicine therapy.”
Line 275-277: Replace “Similarly, many authors report that, in case of allergic patients, they have recorded a decrease in initial elevated concentrations of hematological parameters during long-term treatment due to 277 chronic therapeutic process” with “Similarly, many authors report that in the case of allergic dogs, they have observed a decrease in the initial elevated concentrations of hematological parameters during long-term treatment.”
Line 348-349: Licorice (Glycyrrhiza glabra) is a entire plant and does not itself a precursor to cortisol. Replace “Licorice (Glycyrrhiza glabra) is a natural precursor of cortisol.” with “Glycyrrhiza glabra contains plant constituents that inhibit renal 11β-hydroxysteroid dehydrogenase activity, thereby reducing the conversion of cortisol to cortisone and resulting in high renal levels of cortisol, which is available for binding to mineralocorticoid receptors.”
Line 355-356: Replace “However, its overdosing develops edema.” with “However, too high a dose of Licorice (Glycyrrhiza glabra) can lead to edema, although deglycyrrhizinated versions Licorice are also available.”
356-357: Replace “Licorice contains potassium that causes an imbalance between sodium and potassium that evolve a cardiac arrhythmias and hypertension” with “Licorice contains potassium levels that may cause an imbalance between sodium and potassium leading to cardiac arrhythmias and hypertension”
Line 367-368: Not sure what is meant by “anti-infective” activity? Do you mean antibiotic activity? “In addition to their immunostimulatory effect, they also have anti-infective activity and increase the body's resistance”
Line 390-391: IT appears there are no references for Turmeric (Curcuma longa) listed in the references, and reference (42) is actually for Beta-Glucan. Please provide additional references for Turmeric (Curcuma longa).
The references appear to be formatted in an unusual format as the names are all in capitals and some titles are in italics and others are not, and some journal names are in italics and others not. Please choose an appropriate APA or AMA or MDPI Animals-approved format for references and use that.
Round 2
Reviewer 2 Report
animals-1739603 new revised compared
Once the below changes have been made this reviewer believes the manuscript is ready for publication. It would be good to have a native English speaker go over it after the authors have included the changes I list below. They have been very responsive to suggestions and that is appreciated
Please remember when introducing an abbreviated name for anything in ta research manuscript, the first time it appears spell out the full name and then in parentheses place the abbreviation, and then use the abbreviation from that point on.
Line 7 & 8: Replace “Acupuncture, phytotherapy and nutrition are part of traditional Chinese medi-7 cine, which has been used for many years” with “
Acupuncture, phytotherapy and nutrition are part of traditional Chinese medicine, which has been used for literally hundreds to a few thousand years”
Line 9: Replace “and can be used as main or complementary therapy” with “ and can be used as a primary or complementary therapy”
Line 19: Replace “In case of highly intensive pruritus” with “In the case of highly intensive pruritus”
Line 24: “but can treat the patient as a unit.” with “ “but can treat the patient as a whole.”
Line 46: Replace “through intestinal wall” with “through the intestinal wall”
Line 48: Replace “In dogs, are considered to be the most common allergens beef, pork, chicken and also 48 dairy products” with “In dogs, beef, pork, chicken and dairy products are considered the most common allergens”
Line 62-63: Please remove this sentence as it is redundant: “Allergic diseases have been serious disabilities that affected individuals suffer for the rest of their lives.”
Line 66: Replace “what is unfortunately” with “that is unfortunately”
Line 66-67: ”Replace “with many side effects, that affect” with “with many side effects that affect”
Line 67-68: Replace “conventional 67 medicine therapies” with “conventional Western medicine therapies”
Line 78-79” Replace “a conventional therapy including corticoids, antibiotics and Janus kinase inhibitors.” With “a conventional Western medicine therapy, including corticoids, antibiotics and Janus kinase inhibitors.”
Line 93-94: Replace “microscopic assessment of skin scraping, that confirmed or excluded the presence of parasites” with ““microscopic assessment of the skin scraping that confirmed or excluded the presence of parasites”
Line 96-97: Replace “in all patients by ELISA blood test provided in diagnostic laboratory LABOKLIN s.r.o.” with “in all patients by an ELISA blood test provided by the diagnostic laboratory 96 LABOKLIN s.r.o.”
Line 99-100: Replace “In the first group of patients was performed diagnostics according to principles of traditional Chinese medicine (TCM)” with “In the first group of patients diagnostics was performed according to the principles of traditional Chinese medicine (TCM)”
Line 101: Replace “The treatment was provided by acupuncture and phytotherapy that divide meals into three different groups due to energetic effect of each food (hot, cold and neutral)” with “The treatment was acupuncture and phytotherapy that divides meals into three different groups due to the energetic effects of each food (hot, cold and neutral)”
Line 104-105: Replace “In case of allergic skin manifestacions and ear infections in dogs there is usually excess of Yang energy” with “In case of allergic skin manifestations and ear infections in dogs there is usually an excess of Yang energy”
Line 109-110: Replace “according to its purpose according to its purpose in TCM theory” with “according to its purpose in TCM theory”
Line 117: Replace “according to the TCM in case of allergy” with “according to TCM in case of allergy”
Line 123: Replace “Diagnosis and therapy in the second group” with “Diagnosis and treatment in the second group”
Line 126: Replace “All dogs were fed by hypoallergenic diet” with “All dogs were fed a hypoallergenic diet”
Line 148: Replace “Evaluation of hematological analysis of the first group of dogs” with “Hematological analysis of the first group of dogs”
Line 149: Replace “where the most numerous changes” with “where the largest changes”
Line 154-155, Line 157-158: You do not need to capitalize eosinophils (Eos), leukocytes (Leu) and neutrophils (Neu).
Line 158-159: Do not capitalize “erythrocyte (Ery) and lymphocyte (Lym)
Line 162-163: Replace “with asterisk *” with “with an asterisk *”
Line 162: Replace “Tera (10^ 12) /liter ; G/L, Giga (10^ 9) /liter” with “trillions (10^ 12) /liter ; G/L, billions (10^ 9) /liter”
Line 167-168: Replace “In case of the liver enzyme ALT was the mean value” with “In the case of the liver enzyme ALT the mean value”
Line 170-171: Usually whenever reporting the overall mean you also must include the standard deviation—this is a standard practice for any publication.
Line 176: Replace “when their mean level” with “where their mean level”
Line 177-178: Replace “After therapy were serum levels of pancreatic enzymes adjusted in all observed patients” with “After therapy serum levels of pancreatic enzymes adjusted in all observed patients”
Line 178: Replace “In case of urea (BUN) and creatinine” with “In the case of urea (BUN) and creatinine”
Line 180: Replace “effect on the overall urea parameter” with “effect on the overall urea mean”
Line 107: please include the standard deviation whenever reporting a mean value.
Line 208-209: Replace “but in two cases the number of eosinophils still exceeded and the average of the reference value has been 1.52 x10^ 9/L “ with “but in two cases the number of eosinophils still exceeded the average of the reference value (1.52 209 x10^ 9/L)”
Line 210-211: Replace “In case of leukocytes” with “In the case of leukocytes”
Line 217-218: Replace “Tera (10^ 12) /liter ; G/L, Giga (10^ 9) /liter” with “trillions (10^ 12) /liter ; G/L, billions (10^ 9) /liter”
Line 223: Replace “of Eoz and Leu concentrations” with “of Eos and Leu concentrations”
Line 226-229: Please always include the standard deviation whenever you provide an average value. This is standard practice for any research publication.
Line 232-233: Replace “Because of these animals were temporarily stabilized or still ongoing therapy” with “Because these animals were temporarily stabilized or still receiving ongoing therapy”
Line 234-235: Replace “Increment of pancreatic enzymes (AMS and LPS) was also observed” with “Incremental increases of pancreatic enzymes (AMS and LPS) was also observed”
Line 240: Replace “when their average level (1892 U / L) was increased” with “where their average level (1892 U / L) was increased”
Line 243-244: Replace “Recorded also were an increase in level of urea and creatinine” with “an increase in level of urea and creatinine was also recorded in the level of urea and creatinine”
Line 292: Replace “other therapeutic option used in classical medicine” with “other therapeutic option used in conventional Western medicine”
Line 310: Replace “which may be related with the disease” with “which may be related to the disease”
Line 312” Replace “which confirms that also alternative” with “which confirms that alternative”
Line 326: Replace “patients of second study group” with “patients from the second study group”
Line 329-330: Replace “the second group of dogs had tendency” with “the second group of dogs had a tendency”
Line 333: Replace “an important deal of treatment” with “an important part of treatment”
Line 344: Replace “According to the TCM” with “According to TCM”
Line 347: Replace “the most important is also patient's current energy and health” with “the most important is also the patient's current energy and health”
Line 356: Replace “Another study shows that acupuncture” with “Another study showed that acupuncture”
Line 375: Please place all scientific names for plants such as Glycyrrhiza glabra in italics: Glycyrrhiza glabra
Line 386: Do not capitalize Licorice “although deglycyrrhizinated versions Licorice are also available”
Line 398: Replace “They have immunomodulatory effect, stimulate immune system” with “They have immunomodulatory effects, stimulate the immune system”
Line 412: Again please place scientific names of plant in italics: Turmeric (Curcuma longa)
Line 423-424: Replace “but only temporary so recurrence of the allergic symptoms developed” with “but only temporarily, so recurrence of the allergic symptoms developed
Line 431-441: REpalce “The knowledge obtained in this study research cannot be presented only as unequivocal and comprehensive instruction to solve mentioned problem. The aim of this study was to show and pay attention to the existence of alternative diagnostic and therapeutic methods that have been used in the past and which are becoming nowadays very popular. According to many worldwide researches traditional therapeutic methods, such as acupuncture, phytotherapy and others, which have been used for centuries, are also very beneficial in the treatment of allergies. If these non-invasive methods are used properly, they can significantly improve or even stabilize many health problems and diseases. Alternative options are relatively safe, without risk of side effects and do not harm the body. They can be used separately or in combination with other therapies, and can be used as an independent or complementary therapy for both animals and humans.” with “The observations made in this study cannot be presented as unequivocal and comprehensive instruction in solving the problem of allergy in dogs. The aim of this study was to show and pay attention to the existence of alternative diagnostic and therapeutic treatment methods that have been used in the past and which are becoming nowadays very popular. According to much worldwide research traditional therapeutic methods, such as acupuncture, phytotherapy and others, which have been used for centuries, are also very beneficial in the treatment of allergies. If these non-invasive methods are used properly, they can significantly improve or even stabilize many health problems and diseases with minimal side effects. Alternative options are relatively safe, without risk of side effects, and do not harm the body. They can be used separately or in combination with other therapies, and can be used as an independent or complementary therapy for both animals and humans.”
